# Recombinant *Salmonella gallinarum* (*S. gallinarum*) Vaccine Candidate Expressing Avian Pathogenic *Escherichia coli* Type I Fimbriae Provides Protections against APEC O78 and O161 Serogroups and *S. gallinarum* Infection

**DOI:** 10.3390/vaccines11121778

**Published:** 2023-11-28

**Authors:** Peng Dai, Hucong Wu, Guowei Ding, Juan Fan, Yuhe Li, Shoujun Li, Endong Bao, Yajie Li, Xiaolei Gao, Huifang Li, Chunhong Zhu, Guoqiang Zhu

**Affiliations:** 1Joint Laboratory of International Cooperation on Prevention and Control Technology of Important Animal Diseases and Zoonoses of Jiangsu Higher Education Institutions, Yangzhou University, Yangzhou 225012, China; daipeng@jinyu.com.cn; 2College of Veterinary Medicine, Yangzhou University, Yangzhou 225012, China; 3Yangzhou Uni-Bio Pharmaceutical Co., Ltd., Yangzhou 225008, China; dingguowei@jinyu.com.cn (G.D.); fanjuan@jinyu.com.cn (J.F.); liyuhe@jinyubaoling.com (Y.L.); 4Nei Monggol Animal Disease Control Center, Hohhot 010010, China; wuhucong@yzu.edu.cn; 5Tianjin Ringpu Bio-Technology Co., Ltd., Tianjin 300308, China; shoujun@ringpu.com (S.L.); baoed@ringpu.com (E.B.); yajie@ringpu.com (Y.L.); xiaolei@ringpu.com (X.G.); 6Jiangsu Institute of Poultry Sciences, Yangzhou 225125, China; huif@yzu.edu.cn (H.L.); zhuch_1304428@yzu.edu.cn (C.Z.)

**Keywords:** APEC, type I fimbriae, *Salmonella* vector, safety, protective effect

## Abstract

Avian pathogenic *Escherichia coli* (APEC) is one of the leading pathogens that cause devastating economic losses to the poultry industry. Type I fimbriae are essential adhesion factors of APEC, which can be targeted and developed as a vaccine candidate against multiple APEC serogroups due to their excellent immunogenicity and high homology. In this study, the recombinant strain SG102 was developed by expressing the APEC type I fimbriae gene cluster (*fim*) on the cell surface of an avirulent *Salmonella gallinarum* (*S. gallinarum*) vector strain using a chromosome-plasmid-balanced lethal system. The expression of APEC type I fimbriae was verified by erythrocyte hemagglutination assays and antigen-antibody agglutination tests. In vitro, the level of the SG102 strain adhering to leghorn male hepatoma (LMH) cells was significantly higher than that of the empty plasmid control strain, SG101. At two weeks after oral immunization, the SG102 strain remained detectable in the livers, spleens, and ceca of SG102-immunized chickens, while the SG101 strain was eliminated in SG101-immunized chickens. At 14 days after the secondary immunization with 5 × 10^9^ CFU of the SG102 strain orally, highly antigen-specific humoral and mucosal immune responses against APEC type I fimbriae protein were detected in SG102-immunized chickens, with IgG and secretory IgA (sIgA) concentrations of 221.50 μg/mL and 1.68 μg/mL, respectively. The survival rates of SG102-immunized chickens were 65% (13/20) and 60% (12/20) after challenge with 50 LD_50_ doses of APEC virulent strains O78 and O161 serogroups, respectively. By contrast, 95% (19/20) and 100% (20/20) of SG101-immunized chickens died in challenge studies involving APEC O78 and O161 infections, respectively. In addition, the SG102 strain effectively provided protection against lethal challenges from the virulent *S. gallinarum* strain. These results demonstrate that the SG102 strain, which expresses APEC type I fimbriae, is a promising vaccine candidate against APEC O78 and O161 serogroups as well as *S. gallinarum* infections.

## 1. Introduction

Avian pathogenic *E. coli* (APEC) is a subgroup of extraintestinal pathogenic *E. coli* (ExPEC) that causes colibacillosis, which presents as a range of syndromes including pericarditis, perihepatitis, and meningitis outside of the gastrointestinal tract [1]. The birds afflicted with avian colibacillosis often exhibit extreme lethargy. A reduced water intake is indicative of a grave prognosis. The severely affected individuals exhibit unresponsiveness upon approach and show a lack of reaction to stimuli [2]. APEC is one of the most common bacterial pathogens that infect chickens, causing annual billion-dollar losses to the poultry industry worldwide, especially in antibiotic-free programs [3,4,5]. In “no antibiotics ever” broiler farms, 93% of samples, including litter samples, fecal samples, cloacal swabs, and tracheal swabs, were tested positive for *E. coli*, and 87.4% of all *E. coli*-positive samples were associated with APEC-like strains [5]. The APEC isolates exhibit genetic similarities in virulence genes with human uropathogenic *E.coli* and neonatal meningitis *E.coli*, which could result in the development of urinary tract infections and meningitis in humans [6]. Recently, there have been reports of the potential for APEC to be zoonotic and disseminate multi-drug resistance to humans [7,8,9,10]. Additionally, numerous serogroups of isolates inevitably pose a challenge to controlling APEC infections. Therefore, it is essential to develop an effective vaccine against multi-serogroup APEC infections. 

An ideal vaccine candidate against APEC should provide broad-spectrum protection efficacy for multiple serogroups of APEC strains rather than being limited to a specific serovar. In recent years, multiple antigens, including O-antigen polysaccharides of APEC, have been designed to be expressed in the periplasm of recombinant *Salmonella* strains using chromosome-plasmid-balanced lethal systems [11,12,13]. However, the innate immune responses and biological functions induced by foreign antigens expressed in the periplasm of the *Salmonella* vector are unclear and require further evaluation. Subsequent studies have shown that the levels of immune responses induced by an antigen expressed on the surface of the *Salmonella* vector are significantly higher than those induced by an antigen expressed in the periplasmic compartment [14,15]. Type I fimbria is biosynthesized by APEC strains and is a critical virulence determinant involved in adherence to eukaryotic cells [16,17]. Most APEC strains are capable of expressing type I fimbriae, which they use to adhere to mucosal epithelial cells in the respiratory and digestive tracts of chickens [18,19,20,21,22]. Previous studies have demonstrated the excellent immunogenicity of vaccine candidates based on type I fimbriae [23,24]. Furthermore, structural homologies and common antigenic epitopes among type I fimbriae from different serogroups suggest that a vaccine targeting multi-serogroup APEC strains could be developed [25,26]. 

Fowl typhoid, caused by *S. gallinarum*, is a prevalent disease on poultry farms worldwide as well. Infection with *S. gallinarum* often results in severe complications such as decreased egg production in layers, reduced hatchability rates of eggs, stunted growth of chickens, and elevated mortality rates [27]. Smith induced random mutations in the genome of the *S. gallinarum* SG9 strain, resulting in the development of the commercial strain SG9R. The efficacy of SG9R was demonstrated in preventing fowl typhoid during challenge experiments [28]. However, the SG9R strain still exhibited residual pathogenicity and caused fowl typhoid symptoms in vaccinated birds. Additionally, the presence of mutation sites in the *aceE* and *rafJ* genes within the SG9R strain’s genome raises potential concerns regarding reversion. Additionally, the close relationship between the SG9R strain and *S. gallinarum* isolates from a fowl typhoid outbreak in Belgium has been elucidated through whole-genome sequencing, highlighting significant safety considerations associated with the SG9R strain [29,30]. Consequently, there is an increased demand for *S. gallinarum* vaccine candidates that exhibit enhanced safety profiles suitable for clinical application. 

In a prior study, an avirulent *S. gallinarum* isolate, SG01, was identified and evaluated for its protective efficacy against experimental fowl typhoid [31]. In this study, a recombinant vaccine candidate derived from SG01 expressing APEC type I fimbriae was constructed and tested for adhering abilities in vitro and in vivo. The immune responses and protective effects of the vaccine candidate against APEC O78 and O161 serogroups, as well as *S. gallinarum* infection, were evaluated in challenge studies. 

## 2. Materials and Methods 

### 2.1. Bacterial Strains, Plasmids, and Growth Conditions 

The strains and plasmids utilized in this study are listed in Table 1. Avirulent *S. gallinarum* isolate SG01 and non-type I fimbriae *E. coli* strain (SE5000) were used to integrate the functional recombineering elements [31,32]. *S. gallinarum* and APEC strains were grown in Luria–Bertani (LB) broth or on agar plates at 37 °C. Strains containing pKD46 or pCP20 for the λ-Red recombination assay were incubated at 30 °C. If necessary, chloramphenicol or ampicillin antibiotics were added to the medium at concentrations of 34 or 100 μg/mL, respectively. Diaminopimelic acid (DAP, Zhengzhou JACS Chem Product Co., Ltd., Zhengzhou, China) was added at concentrations of 50 μg/mL for the growth of the *asd* mutant strains in the absence of pYA3342 (*asd*^+^) [33]. 

### 2.2. Chickens 

The one-day-old and thirty-day-old specific-pathogen-free (SPF) white leghorn chickens were supplied from Beijing Boehringer Ingelheim Viton Biotechnology Co., Ltd. (Beijing, China). All chickens were supplied with sterile, purified water and antibiotic-free food. 

### 2.3. Cloning of APEC Type I Fimbriae Operon and Construction of Chromosome-Plasmid Balance Lethal System of SG01 asd Deletion Mutant 

The primers used in this study are listed in Appendix A. Deletion of the *asd* gene was performed via the λ-Red recombination method [34]. The primers P3 and P4 are used to identify the *asd* gene. Homologous recombinant primers (P1-C1 and P2-C2) are designed on the medial side of P3 and P4, in which the sequences displayed in lowercase letters are homologous to the *asd* gene of the SG01 strain, and the sequences displayed in uppercase letters are homologous to the *cat* gene of the pKD3 plasmid. The homologous recombination fragment containing the *cat* gene was amplified using the pKD3 plasmid as a template. Initially, the pKD46 plasmid was transformed into SG01 to construct the recombinant strain named SG01 (pKD46), which contained the pKD46 plasmid. The expression of the recombinases (Exo, Beta, and Gam) was induced by the addition of L-arabinose during the preparation of competent cells for the SG01 (pKD46) strain. Then, the homologous recombination fragment containing the *cat* gene was transformed into the SG01 (pKD46) strain to replace the *asd* gene fragment with the *cat* gene fragment, and the recombinant strain SG01Δ*asd*::*cat* was constructed. The pCP20 plasmid, which expressed FLP flipase to bind to the FRT sites at both ends of the *cat* gene segment, was transformed into the SG01Δ*asd*::*cat* strain. Ultimately, the SG01 strain lacking the *asd* gene fragment, named SG100, was constructed. 

The APEC type I fimbriae operon from the QD2 strain (O78 serogroup) was amplified by PCR with designed restriction sites via corresponding primers and then inserted into expression vectors pBR322 and pYA3342, respectively. The pBR322 empty plasmid and pBR322-APEC *fim* recombinant plasmid were transformed into non-type I fimbriae *E. coli* SE5000, respectively, to construct the recombinant *E. coli* SE5000 strain containing pBR322 or pBR322-APEC *fim*. The pYA3342 empty plasmid and the pYA3342-APEC *fim* recombinant plasmid were transformed into the SG100 strain, respectively. Ultimately, two recombinant *S. gallinarum* vaccine candidates were established: SG101 strian, containing pYA3342, and SG102 strian, containing pYA3342-APEC *fim*. 

### 2.4. Identification of Type I Fimbriae Expression

Erythrocyte hemagglutination assay. *S. gallinarum* strains SG01, SG101, SG102, and SG103, and *E. coli* strains SE5000 (pBR322-APEC *fim*) and SE5000 (pBR322) were cultured in LB broth under 37 °C overnight, and then 1 mL of each culture was centrifuged at 4000 rpm for 10 min. The pellet was suspended in PBS after washing with PBS three times. A total of 2% chicken erythrocyte suspensions were prepared for hemagglutination tests to observe agglutination particles on the glass slide [35,36]. 

Antigen-antibody agglutination test. The APEC *fim*-mediated agglutination test was determined by the rabbit polyclonal anti-serum to APEC FimA. *S. gallinarum* strains SG01, SG101, SG102, and SG103, as well as *E. coli* strains SE5000 (pBR322-APEC *fim*) and SE5000 (pBR322), were cultured in LB broth under 37 °C overnight. The bacterial count was adjusted to 5 × 10^9^ CFU/mL, followed by washing with sterile saline three times before performing the agglutination assay. Then, 10 μL of bacterial suspension was mixed with an equal volume of rabbit polyclonal anti-serum to APEC FimA diluted at a ratio of 1:40 on a glass slide to observe agglutination particles. 

Transmission electron microscopy (TEM) observation of fimbriae structure. The SE5000 (pBR322-APEC *fim*) and control strain SE5000 (pBR322) were cultured in LB broth under 37 °C overnight. Then, 1 mL of each culture was centrifuged at 4000 rpm for 10 min, and the thallus was suspended in PBS after washing with PBS three times. Fimbriae structure was observed by TEM, as previously described [32]. 

### 2.5. Bacterial Adherence and Adherence Inhibition Assays 

Fimbriae-mediated adherence properties of SG101 and SG102 strains were determined by cell adhesion and adherence inhibition assays as previously described [37]. 1 × 10^7^ CFU of bacteria suspended in PBS were added to a monolayer of around 1 × 10^5^ LMH cells in each well of a 96-well plate (Corning Inc., Corning city, NY, USA). After incubating for 1 h, the cell monolayer was washed with PBS three times and subsequently lysed with 0.5% Triton X-100 for 30 min. The lysates were diluted 1:10 in PBS and plated on LB agar plates under 37 °C overnight for the enumeration of adherent bacteria. To perform the adherence inhibition assay, bacteria were suspended in PBS containing 5% D-mannose before incubation with LMH cells. All other procedures are the same as those for bacterial adherence assays [37]. 

### 2.6. Safety and Virulence Assessment 

To assess the safety of SG01-derived strains (SG101 and SG102), the 50% lethal doses (LD_50_) of these strains were evaluated. Eighty SPF white leghorn chickens (one-day-old) were randomly divided into eight groups, with four groups inoculated with the SG101 strain and four groups inoculated with the SG102 strain, each group containing ten chickens. Chickens were orally inoculated with the SG101 or SG102 strain in doses of 1 × 10^8^, 1 × 10^9^, 1 × 10^10^, and 1 × 10^11^ CFU per chicken, respectively. 

The LD_50_ of APEC wild strains (QD2, O78 serogroup, and O161, O161 serogroup) were also evaluated to determine the appropriate challenge dose for subsequent studies. Eighty SPF white leghorn chickens (thirty-day-old) were randomly divided into eight groups, with four groups inoculated with the QD2 strain and four groups inoculated with the O161 strain, each group containing ten chickens. Chickens were inoculated with QD2 or O161 strains at doses of 1 × 10^7^, 1 × 10^8^, 1 × 10^9^, and 1 × 10^10^ CFU, respectively, by the posterior chest air sac route. The LD_50_ was calculated by the Bliss method using IBM SPSS Statistics 25.0 [38]. 

### 2.7. Antigen-Specific Antibodies against Salmonella Vector Strain 

After wing tags were performed, thirty SPF white leghorn chickens (one-day-old) were randomly divided into three groups, each group containing ten chickens: the SG101 group, the SG102 group, and the PBS group. A total of 200 μL of 5% NaHCO_3_ suspensions were used to neutralize stomach acid before inoculations. The SG101 group and the SG102 group were orally inoculated with 5 × 10^9^ CFU of the SG101 or SG102 strain in 200 μL of PBS, respectively, while the PBS group was only orally inoculated with 200 μL of PBS. Serum samples were collected weekly from all chickens until ten weeks post-inoculation. Agglutination reactions against the *Salmonella* vector strain were tested using *Salmonella peg*-expressing agglutination antigens (Patent: CN202010400163.4) [39]. Double dilution methods were used to dilute each serum sample with 0.9% normal saline solution, followed by mixing 10 μL of agglutinations with an equal volume of serum sample on a clean glass plate. Specific antigen-antibody reactions were observed as agglutination particles within 2 min. 

### 2.8. Bacterial Vaccination and Measurements of Body Weight in Chickens 

After wing tags were performed, ninety SPF white leghorn chickens (one-day-old) were randomly divided into three groups, each group containing thirty chickens: the SG101 group, the SG102 group, and the PBS group. A total of 200 μL of a 5% NaHCO_3_ suspension was used to neutralize stomach acid before inoculation. The SG101 group and SG102 group were orally inoculated with 5 × 10^9^ CFU of the SG101 or SG102 strain in 200 μL of PBS, respectively, while the PBS group was only orally inoculated with 200 μL of PBS. After two weeks, the SG101 and SG102 groups received a secondary oral inoculation of 5 × 10^9^ CFU of the respective strains in 200 μL of PBS, while the PBS group was still orally inoculated with 200 μL of PBS. The body weights of a fixed number of 10 chickens from three groups were measured at 1, 14, and 28 days after the initial immunization. 

### 2.9. In Vivo Persistence of Bacteria in Chicken Tissues 

The liver, spleen, and cecum samples were aseptically collected from five chickens in each group at 1, 7, 14, and 21 days after the secondary immunization. The samples were weighed and suspended in 500 μL of PBS for homogenization. Serial dilutions of homogenates (100 μL for each dilution) were inoculated onto brilliant green agar (Hopebio, China) plates for overnight cultivation at 37 °C to recover bacteria. The bacterial count was determined and reported as log_10_ CFU/g. 

### 2.10. Specific Humoral and Mucosal Immune Response 

The humoral immune responses were evaluated by measuring systemic IgG titers by indirect ELISA. Blood samples were collected from a fixed number of 10 chickens at 7, 14, 21, and 28 days after the first immunization via the wing vein. Serum samples were obtained by centrifuging blood samples at 4000 rpm for 5 min as test samples. ELISA plate wells (Greiner Bio-One GmbH, Taufkirchen, Germany) were coated with 100 μL of FimA, the major subunit of type I fimbriae, at a concentration of 0.2 mg/mL and incubated overnight at 4 °C. Then, the ELISA plate wells were washed five times with PBST. A total of 100 μL of each standard, zero control, and test sample were added into wells and incubated for 1 h. After another round of washing with PBST, 100 μL of horseradish peroxidase-conjugated goat anti-chicken IgG (Abcam, Cambridge, UK) at a dilution of 1:10,000 was added into the washed wells and incubated for 1 h. Following another round of washing, 100 μL of tetramethylbenzidine (Solarbio, Beijing, China) was added to the reaction well, incubated in the dark for 0.5 h, and finally terminated with 50 μL of H_2_SO_4_ (2 M). The value of OD_450nm_ was measured using an automated microplate reader (Bio-Rad, Hercules, CA, USA). The readings for each standard, sample, and zero control values were averaged, and the zero control was subtracted from all mean readings. Subsequently, the mean readings were plotted against the concentrations, and a smooth curve was drawn through the points to construct a standard curve using the Graphpad 8.0 software. The IgG concentrations for the test samples were extrapolated from the plotted standard curve. 

The mucosal immune responses were monitored by measuring sIgA levels. Intestinal lavage samples from a fixed number of 10 chickens were collected at 7, 14, 21, and 28 days after the first immunization to determine sIgA titers [40]. Chickens were orally administered a lavage solution consisting of 48.5 mM polyethylene glycol, 40 mM Na_2_SO_4_, 20 mM NaHCO_3_, 10 mM KCl, and 25 mM NaCl (Macklin, Shanghai, China). In addition, 30 min later, 5% pilocarpine nitrate (Macklin, China) dissolved in ultrapure water was administered intramuscularly into chickens by the intramuscular route. Secretions from the chicken cloaca were collected in PBST with a soybean trypsin inhibitor (Roche, Basel, Switzerland) and centrifuged at 12,000 rpm under 4 °C for 10 min. The supernatant was collected, and sIgA titers were determined by the chicken IgA ELISA Quantitation Kits (Abcam, UK). The ELISA plate wells were coated with 100 μL of FimA at a concentration of 0.2 mg/mL and incubated overnight at 4 °C. After washing with PBST five times, wells were reacted with intestinal lavage samples at dilutions of 1:100 for 1 h, followed by reaction with horseradish peroxidase-conjugated goat anti-chicken IgA at dilutions of 1:50,000 for 1 h. Then, 100 μL of tetramethylbenzidine (Solarbio, Beijing, China) was added to the reaction well, incubated in the dark for 0.5 h, and finally terminated with 50 μL of H_2_SO_4_ (2 M). The concentration of sIgA was quantified following the same procedure as IgG quantitation. 

### 2.11. Protective Effects against APEC O78 and O161 Serogroups Challenge 

After wing tags were performed, one hundred and twenty SPF white leghorn chickens (one-day-old) were randomly divided into six groups (Groups A, B, C, D, E, and F), each group containing 20 chickens. 200 μL of a 5% NaHCO_3_ suspension was used to neutralize stomach acid before inoculation. Groups A and B were orally immunized with 5 × 10^9^ CFU of the SG101 strain in 200 μL of PBS. Groups C and D were orally immunized with 5 × 10^9^ CFU of the SG102 strain in 200 μL of PBS. Groups E and F served as controls by receiving only 200 μL of PBS. Two weeks after initial inoculation, Groups A and B received a secondary oral immunization of 5 × 10^9^ CFU of the SG101 strain in 200 μL of PBS, while Groups C and D received a secondary oral immunization of 5 × 10^9^ CFU of the SG102 strain in 200 μL of PBS. Meanwhile, Groups E and F continued to receive an oral inoculation with 200 μL of PBS. Subsequently, 15 days after the secondary immunization, groups A, C, and E were challenged with a 50 LD_50_ dose of APEC QD2 strain (O78 serogroup) by the posterior chest air sac route, while groups B, D, and F were challenged with a 50 LD_50_ dose of APEC O161 strain (O161 serogroup) by the same route. The number of dead chickens was recorded within 14 days. 

### 2.12. Protective Effects against S. gallinarum Challenge 

After wing tags were performed, sixty SPF white leghorn chickens (one-day-old) were randomly divided into three groups, each group containing 20 chickens: the SG101 group, the SG102 group, and the challenge group. A total of 200 μL of a 5% NaHCO_3_ suspension was used to neutralize stomach acid before inoculation. The SG101 and SG102 groups were orally administered with 5 × 10^9^ CFU of the SG101 or SG102 strain in 200 μL of PBS, respectively. After 4 weeks, all chickens were challenged with 2 × 10^10^ CFU of the *S. gallinarum* virulent strain U20 via the intraperitoneal route. The number of dead chickens was recorded within 14 days. 

### 2.13. Statistical Analysis 

All data were recorded as the mean ± standard error of the mean (SEM). The statistical analyses were performed with SPSS 25.0. Unless otherwise specified, an unpaired Student’s t test was used to analyze significant differences between the SG101 group and the SG102 group. 

## 3. Results 

### 3.1. Identifications of Asd Deletion Mutant and Recombinant Plasmids 

Identifications of the *asd* deletion mutant and recombinant plasmids are shown in Figure 1. SG01Δ*asd* mutant (SG100), pBR322-APEC *fim*, and pYA3342-APEC *fim* recombinant plasmids were obtained. After transformations, SE5000, containing pBR322-APEC *fim*, and SG102, containing pYA3342-APEC *fim*, were subsequently obtained. 

### 3.2. Verifications of APEC Type I Fimbriae Expression 

Verifications of APEC type I fimbriae expression on the SE5000 strain containing pBR322-APEC *fim* were shown in Figure 2. Hemagglutination tests showed SE5000 (pBR322-APEC *fim*) could agglutinate with 2% chicken erythrocyte suspensions, while the SE5000 strain containing the empty vector pBR322 did not (Figure 2A). In addition, the SE5000 (pBR322-APEC *fim*) instead of the SE5000 (pBR322) strain could react with polyclonal anti-APEC FimA antibodies (Figure 2B). TEM observations were performed for the further confirmation of type I fimbriae expressions. Significant fimbria structures were found on the surface of SE5000 (pBR322-APEC *fim*) but not on the control strain SE5000 (pBR322) (Figure 2C). 

APEC type I fimbriae expression was verified on recombinant *S. gallinarum* strains as well. As shown in Table 2, the SG102 strain containing pYA3342-APEC *fim* agglutinated with 2% chicken erythrocyte suspensions and polyclonal anti-APEC FimA antibodies, while the SG101 strain containing pYA3342 and the SG103 strain containing pYA3342-APEC *fimA* did not agglutinate with 2% chicken erythrocyte suspensions and polyclonal anti-APEC FimA antibodies. 

### 3.3. Bacterial Adherence and Adherence Inhibition In Vitro 

LMH cells were used for bacterial adherence and adherence inhibition assays in vitro. The results showed that about 3.5 × 10^5^ CFU of the SG102 strain adhered to LMH cells, which was 4.2-fold higher (*p* < 0.05) compared to the control strain SG101 containing the empty plasmid pYA3342. Additionally, adhesion levels of the SG102 and SG101 strains were not significantly different after incubating with 5% D-mannose (Figure 3). The bacterial adherence and adherence inhibition assays were repeated three times, respectively. 

### 3.4. Virulence of Vaccine Candidate and Challenge Strains 

The virulence of the vaccine candidate was tested in one-day-old SPF white leghorn chickens for safety evaluation, while the virulence of the two APEC wild strains was assessed in thirty-day-old SPF white leghorn chickens to determine the challenge dose in subsequent assays. As shown in Table 3, both SG101 and SG102 strains had an LD_50_ of more than 1 × 10^11^ CFU in one-day-old SPF white leghorn chickens. Additionally, the LD_50_ values for QD2 strain (O78 serogroup) and O161 strain (O161 serogroup) were 4.87 × 10^8^ CFU and 5.41 × 10^7^ CFU, respectively. 

### 3.5. Average Anti-Salmonella Peg Antibody Agglutination Titers of Chickens with Different Immune Periods 

The weekly measurement of average agglutination titers of anti-*Salmonella* peg antibodies was conducted in chickens immunized with either the SG101 or SG102 strain. Negative agglutination reactions were identified in the serum samples collected from the PBS group each week after inoculation. Positive agglutination reactions were first detected at 3 weeks post-inoculation (3 wpi). A 1:1 titer was observed in three out of ten (3/10) SG101-immunized chickens and in five out of ten (5/10) SG102-immunized chickens, respectively (Appendix A). The average agglutination titers of anti-*Salmonella peg* antibodies increased significantly for the SG101 and SG102 groups at 4 wpi. Furthermore, chickens immunized with the SG101 strain experienced a decline in agglutination titers starting at 8 wpi. By contrast, those immunized with the SG102 strain showed this trend one week later (Appendix A). At 10 wpi, specific agglutination titers were still detectable in eight out of ten (8/10) SG101-immunized chickens and all ten (10/10) SG102-immunized chickens (Appendix A). It is worth noting that the average anti-*Salmonella peg* antibody agglutination titer of the SG102 group was significantly higher than that of the SG101 group at 5, 6, 7, 8, 9, and 10 wpi (Appendix A). 

### 3.6. Body Weight Measurement and Bacterial Persistence in Chickens after Immunizations 

The mean body weights of chickens from the immunized groups and the PBS group at 1, 14, and 28 days after the first immunization are shown in Figure 4B. There was no statistically significant difference in mean body weights among the two immunized and the PBS groups. Furthermore, livers, spleens, and ceca samples were collected from each group for bacterial recovery. *Salmonella* was not detected in any of the samples collected from chickens in the PBS group. Four out of five (4/5) SG101-immunized chickens and all five (5/5) SG102-immunized chickens tested positive for the corresponding challenge strains in spleen samples at 1 day post-inoculation (1 dpi). The bacterial loads of the SG102 strain were significantly higher than those of the SG101 strain in livers, spleens, and ceca at 1, 7, and 14 dpi (*p* < 0.05, Figure 4C,D). At 14 dpi, the SG101 strain was eliminated in the SG101 group, while the SG102 strain remained detectable in livers, spleens, and ceca from the SG102 group. Additionally, the SG102 strain was no longer detected in SG102-immunized chickens at 21 dpi (Figure 4E). 

### 3.7. Specific Immune Responses against APEC FimA

The serum IgG levels induced by the vaccine candidate were detected by the indirect-ELISA assay. At 14, 21, and 28 dpi, a significant increase in serum IgG against APEC FimA was observed in the SG102 group compared to the SG101 group (Figure 5B). At 14 dpi, the mean IgG concentration of the SG102 group was 40.80 μg/mL, which was 2.5-fold higher (*p* < 0.05) compared to the SG101 group (16.42 μg/mL). Furthermore, the SG102 group (221.50 μg/mL) showed a 7.6-fold increase (*p* < 0.05) in mean IgG concentrations compared to the SG101 group (29.14 μg/mL) at 28 dpi (Figure 5B). 

SIgA concentrations were detected using an indirect-ELISA assay as well. SIgA levels in both the SG101 and SG102 groups were similar to those of the PBS group at 7 and 14 dpi. However, a 1.6-fold and 1.9-fold elevation in mean sIgA concentrations was observed for the SG102 group compared to the SG101 group at 21 and 28 dpi, respectively (Figure 5C).

### 3.8. Protective Effects against APEC Infections 

To assess the protective effects of the vaccine candidate against APEC infection, chickens were inoculated orally with SG101, SG102, or PBS twice, followed by a challenge with 50 LD_50_ doses of APEC strain QD2 (O78 serogroup). The survival rates of chickens from the three groups are shown in Figure 6B. All the chickens (20/20) from the challenge group and 95% (19/20) of the chickens from the SG101 group died within 5 days of the challenge. By contrast, 65% of SG102-immunized chickens (13/20) survived during the observation period, which was significantly higher than the SG101-immunized chickens (*p* < 0.05, Figure 6B). Cardiac and hepatic pathological changes in necropsied birds are shown in Figure 6B. Significant pericarditis pathological changes and livers covered with cellulosic exudates were identified in chickens inoculated with SG101 and PBS, while mild but detectable pericarditis was found in the SG102 group.

The cross-protection effect against the APEC O161 serogroup was also evaluated in young chickens. As shown in Figure 6C, all chickens (20/20) in the SG101 group died within 6 days of challenge, while the survival rate of the SG102 group (60%) was significantly higher than the SG101 group (*p* < 0.05, Figure 6C). 

### 3.9. Protective Effect against S. gallinarum Challenge 

Chickens from SG101, SG102, and the challenge group were immunized once and then challenged with a lethal dose of virulent *S. gallinarum* strain U20 after to assess their protective effect against fowl typhoid. The survival rates of chickens are shown in Appendix A. All of the SG102-immunized chickens survived after the challenge, while 70% of the chickens from the challenge group died during the observation period. It is worth noting that two out of twenty (2/20) SG101-immunized chickens died after the challenge (Appendix A). Pathological changes in the liver and spleen were also observed. A large number of liver necrotic foci and spleen necrotic foci were observed in the challenge group, while only mild hepatosplenomegaly was seen in the SG101 and SG102 groups (Appendix A). 

## 4. Discussion

A high prevalence of APEC strains was found in both developing and developed countries [4,5,41,42,43]. Previous studies have shown that APEC infections are most frequently associated with the serogroups O1, O2, and O78 in poultry farms. However, recent genome sequencing analysis of APEC isolated from Chinese poultry farms has revealed that the prevalence of the O78 serogroup (35.46%) is higher than that of the O2 (6.38%) and O1 (4.26%) serogroups [41]. Moreover, a diverse array of less common serogroups, such as O8, O18, O24, O35, O53, O65, O109, O111, O120, O145, and O161, have been reported in recent years [41,44,45,46]. The significant threat posed by APEC to global poultry production results in substantial economic losses, highlighting the imperative of developing vaccines that can provide robust protection against a broad spectrum of APEC strains. 

Less-virulent *Salmonella* has been shown to effectively induce robust humoral, mucosal, and cell-mediated immune responses as a live vector [47]. The target antigen can be stably expressed by inserting it into an expression plasmid that carries the complementary nutritional deficiency gene of the bacterial vector based on the chromosome–plasmid balanced lethal system [33]. Less-virulent *Salmonella* has been developed as a live vector to deliver multiple foreign antigens, including *E. coli* fimbria, which could stimulate mucosal and systemic Ab- and cell-mediated immune responses in chickens [14,15]. SG01, an avirulent *S. gallinarum* strain, was isolated and showed excellent safety in 1-day-old SPF chickens in the prior study [31]. In this study, the recombinant SG102 strain that expresses, displays, and delivers type I fimbriae on the surface of the SG01 strain based on the chromosome-plasmid balanced lethal system was developed as a vaccine candidate to evaluate its protective effects against the APEC virulent strain challenge. No clinical symptoms or pathological changes were observed in the immunized chickens during the observation period after oral inoculation of different doses of the SG102 strain in one-day-old chickens. Additionally, there was no significant difference in mean daily gain between the immunized groups and the PBS group (Figure 4B). These results indicate that SG102 is a safe vaccine candidate.

A comprehensive understanding of the mechanisms underlying induced immune protection is pivotal for assessing the efficacy of vaccine candidates. Type I fimbria is biosynthesized by APEC strains and is a critical virulence determinant involved in adherence to eukaryotic cells [19,22]. FimH, the adhesin portion of type I fimbriae, mediates the adherence to host cells through specific combinations with mannose receptors (MR), and the adherence can be inhibited by D-mannose [36]. In addition to playing an essential role in host adherence, FimH could activate innate immune responses in vitro and in vivo [48,49]. Furthermore, FimH-mediated mucosal adjuvant effects on the induction of specific immune responses were determined via intranasal administration [50]. Previous studies have shown that multivalent subunit vaccine candidates based on APEC type I fimbriae are effective to a certain extent against challenges from APEC O1, O2, and O78 serogroups [3,24]. However, there have been no subsequent reports or applications of these vaccines [3]. Further consideration is needed for short-term immunity periods and low levels of immunity induction for multivalent subunit vaccines with mixed antigens [51,52]. In this study, by expressing and displaying APEC type I fimbriae on the surface of the vector strain, the in vivo persistence time of the SG102 strain in the liver, spleen, and cecum from SG102-immunized chickens was at least 2 weeks, which is longer than that of the SG101 strain. Bacterial loads of the SG102 strain in livers, spleens, and ceca were significantly higher than those of the SG101 strain (Figure 4C,D). The prolonged persistence of SG102 is a distinctive feature that enhances its colonization levels and adherence capabilities, thereby contributing to heightened protective effects. Additionally, the level of the SG102 strain adhering to LMH cells was significantly higher than that of the SG101 strain in vitro. These results demonstrated the excellent adhering abilities of APEC type I fimbriae, as previous studies have shown [19,21]. Moreover, specific immune responses against the avirulent *Salmonella* vector strain were identified through continuous monitoring in vivo. The antigen-specific antibody titers produced by the SG101 or SG102 strain could still be determined using *Salmonella peg*-expressing agglutination antigens at 10 weeks post-inoculation, demonstrating that long-lasting immune responses can be induced by the avirulent *Salmonella* vector (Appendix A). Further analysis showed that the average anti-*Salmonella peg* antibody agglutination titer of the SG102 group was significantly higher than that of the SG101 group at 5, 6, 7, 8, 9, and 10 weeks post-inoculation (Appendix A). This may be due to the long persistence time of the SG102 strain, mediated by APEC type I fimbriae. 

Oral immunization could induce systemic and mucosal immune responses. In this study, levels of systemic IgG and mucosal sIgA were significantly enhanced against type I fimbriae-specific antigens after oral immunization with the SG102 strain (Figure 5). FimH, the tip adhesin of type I fimbriae, is a novel ligand for toll-like receptor 4 (TLR4) and functions as a mucosal adjuvant while inducing activation of dendritic cells (DCs) [48,50]. Therefore, *fimH*-mediated mucosal adjuvant and immune activations may contribute to the immune protective effect of type I fimbriae against APEC strains. In the challenge assay, the SG102 group exhibited a significantly higher survival rate (65%) compared to the SG101 group (5%) (*p* < 0.05) after being challenged with 50 LD_50_ doses of the APEC O78 serogroup strain. Pathological changes of pericarditis were significantly milder in chickens from the SG102 group compared to those from the SG101 group (Figure 6). The cross-protection effect against the APEC O161 serogroup was also evaluated in one-day-old SPF chickens. After challenge with 50 LD_50_ doses of the APEC O161 serogroup strain, the mortality rate in the SG102 group was significantly lower than that of the SG101 group (Figure 6). These results indicate that SG102, which expresses APEC type I fimbriae, can provide efficient protective effects against APEC O78 and O161 serogroups. A previous study demonstrated the protective efficacy of the *Salmonella* vector strain SG01 against fowl typhoid [31]. This study further showed that both SG101 and SG102 strains derived from the SG01 strain induced excellent protective effects against the virulent *S. gallinarum* challenge (Appendix A). 

In conclusion, we constructed the vaccine candidate SG102 derived from an avirulent *S. gallinarum* strain that provided potent protection against APEC O78 and O161 serogroups as well as *S. gallinarum* infections. These results demonstrate that SG102 is a promising candidate with the potential to significantly improve the specific immune response and economic impact of APEC infection on the poultry industry. Further studies are necessary to evaluate the protective effects of the vaccine candidate against other dominant APEC serogroups, such as O1 and O2, in flocks of varying breeds. In addition, the protective efficacy of SG102 against APEC O78 and O161 serogroup infections should be further enhanced in subsequent studies to optimize its clinical applicability. 

## Figures and Tables

**Figure 1 vaccines-11-01778-f001:**
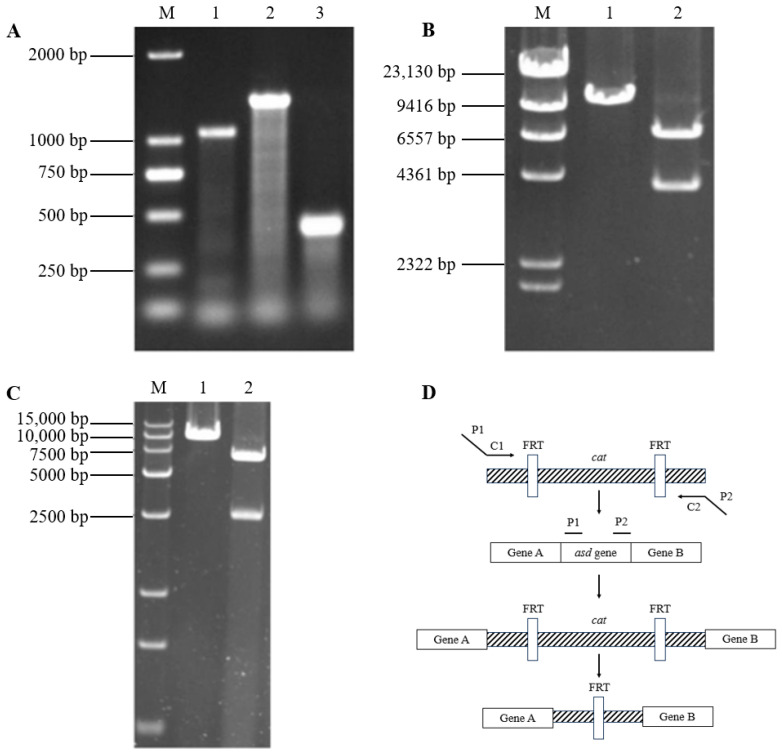
Identifications of the *asd* deletion mutant and recombinant plasmids. (**A**) Identification of the *asd* deletion mutant, DL2000 DNA Marker, lane 1: SG01 strain, lane 2: SG01Δ*asd*::*cat* strain, lane 3: SG01Δ*asd* strain. (**B**) Identification of the pBR322-APEC *fim*, λ-Hind III digest DNA Marker, lane 1: pBR322-APEC *fim* digested by restriction enzyme *Sal* I, lane 2: pBR322-APEC *fim* digested by restriction enzymes *Sal* I and *Nhe* I. (**C**) Identification of the pYA3342-APEC *fim*, DL15000 DNA Marker, lane 1: pYA3342-APEC *fim* digested by restriction enzyme *Sal* I, lane 2: pYA3342-APEC *fim* digested by restriction enzymes *Sal* I and *Nco* I. (**D**) Illustration of *asd* gene deletion using the λ-Red recombination method.

**Figure 2 vaccines-11-01778-f002:**
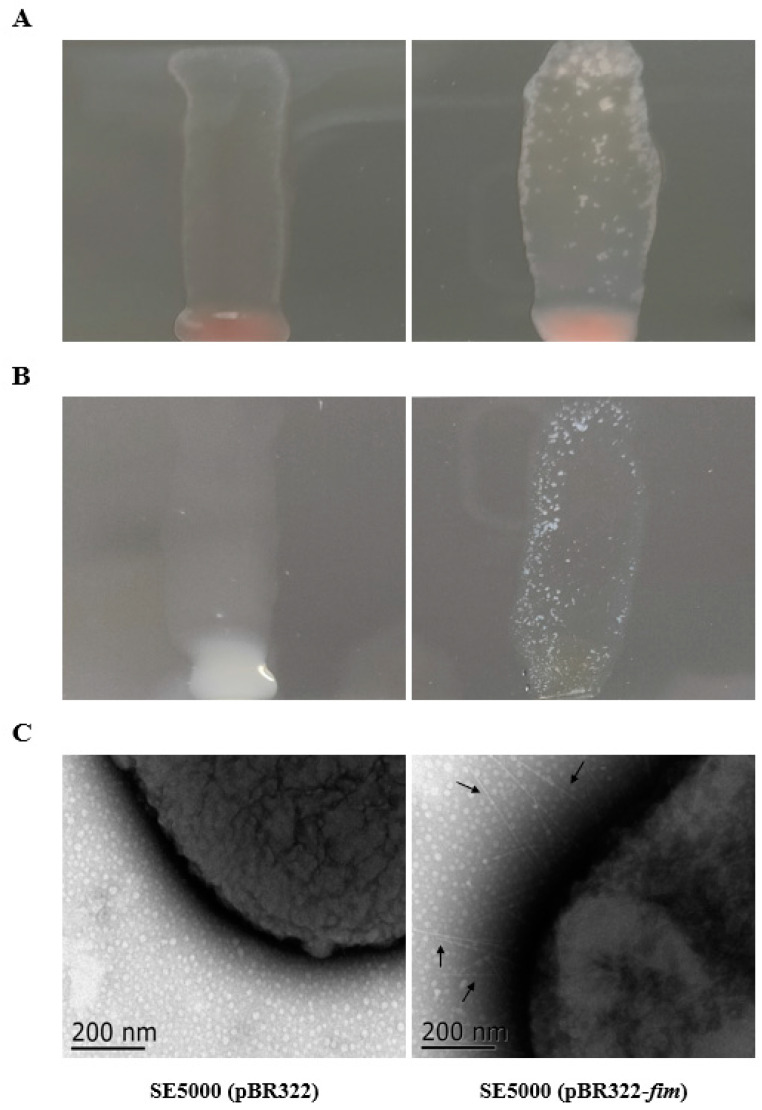
Identification of type I fimbriae expressions. (**A**) MSHA tests. SE5000 (pBR322-APEC *fim*) agglutinated with 2% chicken erythrocyte suspensions, while SE5000 (pBR322) did not. (**B**) Agglutination tests. SE5000 (pBR322-APEC *fim*) agglutinated with 1:40 diluted polyclonal anti-FimA antibodies, while SE5000 (pBR322) did not. (**C**) TEM of fimbriae structures. SE5000 (pBR322-APEC *fim*) showed significant fimbriae structures (black arrows), while SE5000 (pBR322) did not.

**Figure 3 vaccines-11-01778-f003:**
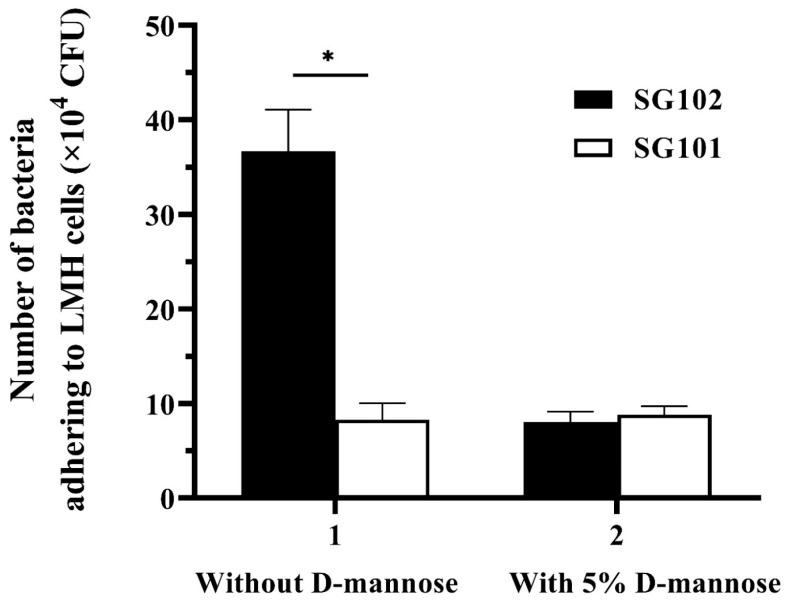
Adherence properties of recombinant *S. gallinarum* strains. Black column: The number of the SG102 strain adhering to LMH cells. White column: The number of the SG101 strain adhering to LMH cells. * *p* < 0.05.

**Figure 4 vaccines-11-01778-f004:**
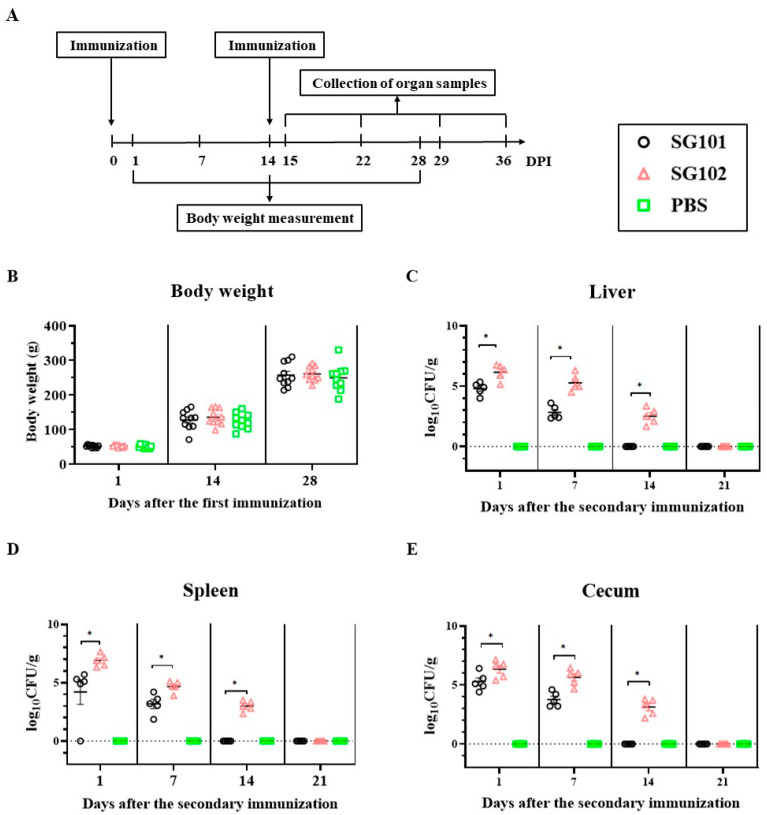
Body weight measurement and bacterial persistence in chickens. (**A**) Chickens were immunized 2 times by SG101, SG102, and PBS suspensions, respectively. The body weights were measured at 1, 14, and 28 days after the first immunization, and organ samples were collected at 15, 22, 29, and 36 days after the first immunization. (**B**) Changes in mean body weights of chickens after immunization. (**C**) Bacterial colonization in chicken livers. (**D**) Bacterial colonization in chicken spleens. (**E**) Bacterial colonization in chicken ceca. The number of bacteria was determined and expressed as log_10_ CFU/g. * *p* < 0.05.

**Figure 5 vaccines-11-01778-f005:**
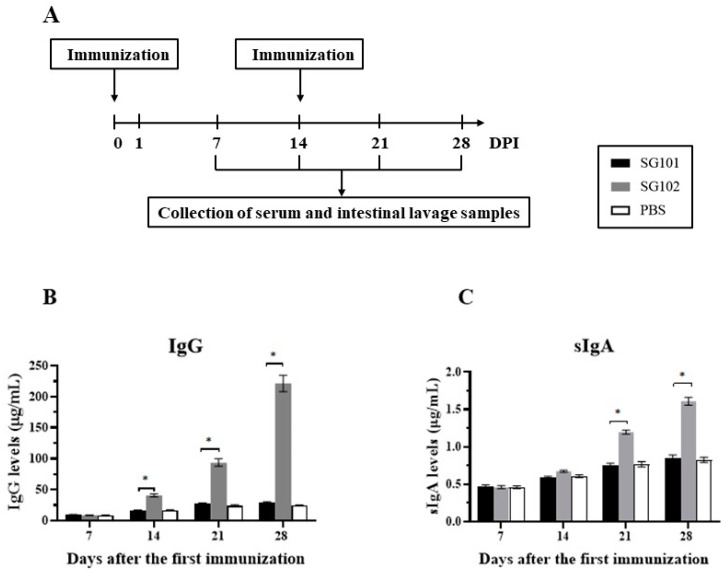
Antigen-specific immune response in chickens. (**A**) Chickens were immunized 2 times by SG101, SG102, and PBS suspensions, respectively. The serum and intestinal lavage samples were collected at 7, 14, 21, and 28 days after the first inoculation. (**B**) Serum IgG levels (μg/mL) by indirect-ELISA. (**C**) Secretory IgA levels (μg/mL) by indirect-ELISA. * *p* < 0.05.

**Figure 6 vaccines-11-01778-f006:**
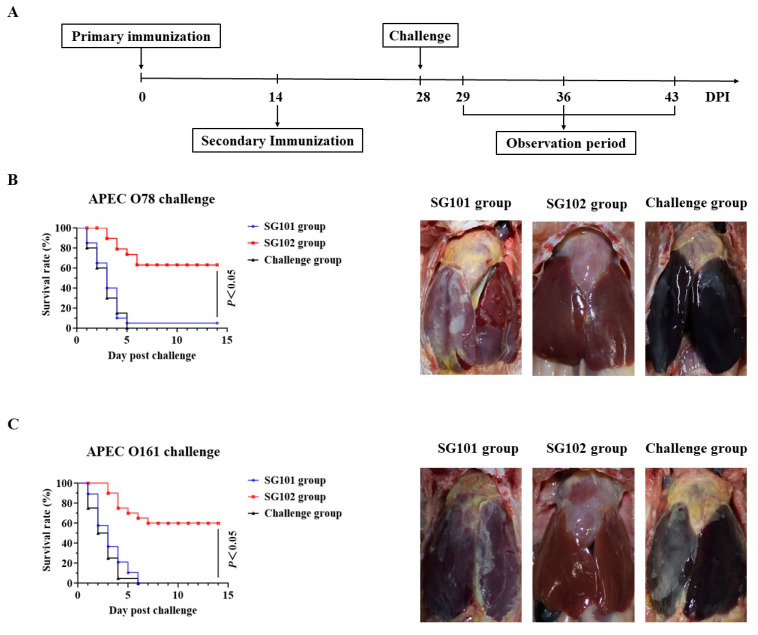
Protective effects against APEC O78 and O161 serogroups infections. (**A**) Chickens were immunized 2 times by SG101, SG102, or PBS suspensions. Fourteen days after the secondary immunization, all chickens were challenged with the APEC QD2 strain (O78 serogroup) or O161 strain (O161 serogroup). Chicken deaths were recorded daily during the 14 days of observation period. (**B**) The survival rate for the SG101, SG102, and PBS inoculated groups after challenge with the QD2 strain and pathological changes of the heart and liver in chickens after challenge with the QD2 strain. (**C**) The survival rate for the SG101, SG102, and PBS inoculated groups after challenge with the O161 strain, and pathological changes of the heart and liver in chickens after challenge with the O161 strain. The differences in survival rates between SG101 and SG102 groups were determined by the log-rank sum test.

**Table 1 vaccines-11-01778-t001:** Strains and plasmids used in this study.

Strains and Plasmids	Desciption	Source
Strains		
SG01	*S. gallinarum* isolate with deficiency of *fimH*	[31]
SG100	Δ*asd* mutant of SG01	This study
SG101	SG100 containing pYA3342	This study
SG102	SG100 containing recombinant pYA3342 expressing APEC type I fimbriae operon	This study
SG103	SG100 containing recombinant pYA3342 expressing APEC *fimA*	Lab stock
U20	*S. gallinarum*, virulent strain	[31]
DH5αΔ*asd*	Δ*asd* mutant of DH5α	Lab stock
QD2	APEC O78 serotype, isolated from chickens	Song Gao
O161	APEC O161 serotype, isolated from chickens	Song Gao
SE5000	*E. coli*, genetic engineering strain without fimbriae	Lab stock
SE5000 (pBR322-APEC *fim*)	SE5000 containing recombinant pBR322 expressing APEC type I fimbriae operon; growing in LB broth with Amp^R^	This study
Plasmids		
pBR322	*expression vector*; Amp^R^, Tc^R^	Lab stock
pBR322-APEC *fim*	pBR322 derivative containing APEC type I fimbriae operon cluster; Amp^R^	This study
pYA3342	*asd*^+^ vector, pBR ori	Dieter M. Schifferli
pYA3342-APEC *fim*	pYA3342 derivative containing APEC type I fimbriae operon	This study
pKD3	*oriR6Kγ*; Amp^R^, Cm^R^	[34]
pKD46	P_BAD_-*gam*-*beta*-*exo oriR101 repA101*^LS^; Amp^R^	[34]
pCP20	cI857 λP_R_ flp *oripSC101^ts^*; Amp^R^, Cm^R^	[34]

**Table 2 vaccines-11-01778-t002:** Verifications of APEC type I fimbriae expression.

Strain	Erythrocyte Hemagglutination Test ^a^	Antigen-Antibody Agglutination Test ^b^
SG01	− ^c^	−
SG101	−	−
SG102	+ ^d^	+
SG103	−	−

^a^ Erythrocyte hemagglutination test was performed using 2% chicken erythrocyte suspensions. ^b^ Antigen-antibody agglutination test was determined by polyclonal anti-APEC FimA antibodies. ^c^ Negative reaction. ^d^ Positive reaction.

**Table 3 vaccines-11-01778-t003:** The LD_50_ of *S. gallinarum* derivative strains (SG101 and SG102) and APEC wild strains (QD2 and O161) in 1-day-old or 30-day-old SPF chickens.

Strains	Inoculation Route	Chicken Age	Inoculation Dose (CFU)	Number of Deaths/Total Numbers of Chickens	LD_50_ (CFU)
SG101	Oral inoculation	1-day-old	1 × 10^8^	0/10	>1 × 10^11^
			1 × 10^9^	0/10	
			1 × 10^10^	0/10	
			1 × 10^11^	0/10	
SG102	Oral inoculation	1-day-old	1 × 10^8^	0/10	>1 × 10^11^
			1 × 10^9^	0/10	
			1 × 10^10^	0/10	
			1 × 10^11^	0/10	
QD2	posterior chest air sac injection	30-day-old	1 × 10^7^	0/10	4.87 × 10^8^
			1 × 10^8^	2/10	
			1 × 10^9^	6/10	
			1 × 10^10^	10/10	
O161	posterior chest air sac injection	30-day-old	1 × 10^7^	3/10	5.41 × 10^7^
			1 × 10^8^	5/10	
			1 × 10^9^	9/10	
			1 × 10^10^	10/10	

## Data Availability

The datasets used and/or analyzed during the current study are available from the corresponding author upon reasonable request.

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
