# Peer review of "Recombinant Salmonella gallinarum (S. gallinarum) Vaccine Candidate Expressing Avian Pathogenic Escherichia coli Type I Fimbriae Provides Protections against APEC O78 and O161 Serogroups and S. gallinarum Infection"

_vaccines, 2023, doi:10.3390/vaccines11121778_

Round 1

Reviewer 1 Report

Comments and Suggestions for Authors

 Title: Recombinant S. Gallinarum vaccine candidate expressing avian pathogenic Escherichia coli type I fimbriae provides protections against APEC O78 and O161 serogroups and S. Gallinarum infection

 General comments

In this study, the authors provide a valuable, interesting and significant study on vaccination of chicken against two important bacterial infections that inducing drastic economic loss in poultry industry; salmonellosis and colibacillosis. The study is well-written regarding English edition, methods and results are clearly and fully described. Protective efficacy was supported by many experiments. However, the study needs some revisions to improve the quality and to be approved for publication as will described in below.

 Whole manuscript

- Word “Gallinarum” should be changed to “gallinarum” and italicized in whole the manuscript including title, texts and figures or figure legends and tables.

- The citation and references style are different than mdpi vaccine-journal style.

 Introduction

- Information on clinical signs of avian colibacilosis and salmonella gallinarum infection should be added in introduction.

- Mechanism of induced immunoprotection of ideal vaccine candidate against investigated infections should be explained in the introduction.

- Previous vaccination trials on avian colibacilosis and salmonella gallinarum infection should be described.

- Information on the current status of commercial used vaccine or control strategies against such infections should be reported.    

 Materials and Methods

- Information on method of blood collection and serum preparation should be added.

- Lines 128, 174, 175 have words with different font size.

- Regarding vaccination regime, how many trials was conducted and if only one trial, the authors should defend this point to guarantee the reproducibility.

 Discussion

The section of discussion is very poor and need addition of more information on the importance of developing vaccines against such infections, mechanism of induced immunoprotection, superiority of SG102 as a vaccine candidate, utility of use against both APEC O78 and O161 serogroups as well as Salmonella gallinarum infections, prospects to improve the survival of chicken in the future studies, efficacy against in case of different type of chickens……   

Comments on the Quality of English Language

The manuscript needs minor English revisions.

Author Response

Response to Reviewer 1 Comments

1. Summary

Thank you for dedicating your valuable time to reviewing this manuscript. Please find the detailed responses below and the corresponding revisions/corrections highlighted/in track changes in the re-submitted files.

2. Questions for General Evaluation Reviewer’s Evaluation

Does the introduction provide sufficient background and include all relevant references? Can be improved

Are all the cited references relevant to the research?

Can be improved Is the research design appropriate? Can be improved

Are the methods adequately described? Can be improved

Are the results clearly presented? Yes

Are the conclusions supported by the results? Can be improved

3. Point-by-point response to Comments and Suggestions

Comments 1: Word “Gallinarum” should be changed to “gallinarum” and italicized in whole the manuscript including title, texts and figures or figure legends and tables.

Response 1: Thank you for pointing this out. We agree with this comment. Therefore, we have modified “Gallinarum” to “gallinarum” in the revised manuscript.

Comments 2: The citation and references style are different than mdpi vaccine-journal style.

Response 2: Agree. We have changed the citation and references style according to the mdpi vaccine-journal style in the revised manuscript.

Comments 3: Introduction section. Information on clinical signs of avian colibacilosis and Salmonella gallinarum infection should be added in introduction.

Response 3: Thank you for pointing this out. We have added the relevant information about clinical signs of avian colibacilosis and Salmonella gallinarum infection in the introduction. “The birds afflicted with avian colibacillosis often exhibit extreme lethargy. A reduced water intake is indicative of a grave prognosis. The severely affected individuals exhibit unresponsiveness upon approach and show a lack of reaction to stimuli.” (Lines 45-48) “Fowl typhoid, caused by S. gallinarum, is a prevalent disease in poultry farms worldwide as well. Infection with S. gallinarum often results in severe complications such as decreased egg production in layers, reduced hatchability rates of eggs, stunted growth of chickens, and elevated mortality rates.” (Lines 78-81)

Comments 4: Introduction section. Mechanism of induced immune protection of ideal vaccine candidate against investigated infections should be explained in the introduction.

Response 4: Agree. We have added the mechanism of induced immune protection by an ideal vaccine candidate against avian colibacilosis infection in the revised manuscript. “An ideal vaccine candidate against APEC should provide broad-spectrum protection efficacy for multiple serogroups of APEC strains, rather than being limited to a specific serovar.” (Lines 60-61) “Type I fimbria is biosynthesized by APEC strains and is a critical virulence determinant involved in adherence to eukaryotic cells. Most APEC strains are capable of expressing type I fimbriae, which they use to adhere to mucosal epithelial cells in the respiratory and digestive tracts of chickens. Previous studies have demonstrated excellent immunogenicity of vaccine candidates based on type I fimbriae. Furthermore, structural homologies and common antigenic epitopes among type I fimbriae from different serogroups suggest that a vaccine targeting multi-serogroup APEC strains could be developed.” (Lines 69-77)

Comments 5: Introduction section. Previous vaccination trials on avian colibacilosis and Salmonella gallinarum infection should be described.

Response 5: The expression of gratitude is extended to you for bringing this matter to my attention. We have added the previous vaccination trials on colibacilosis and S. gallinarum infection in the revised manuscript. “In recent years, multiple antigens, including O-antigen polysaccharides of APEC, have been designed to be expressed in the periplasm of recombinant Salmonella strains using chromosome-plasmid balanced lethal systems. However, the innate immune responses and biological functions induced by foreign antigens expressed in the periplasm of the Salmonella vector are unclear and require further evaluation. Subsequent studies have shown that levels of immune responses induced by antigen expressed on the surface of the Salmonella vector are significantly higher than those induced by an antigen expressed in the periplasmic compartment.” (Lines 62-69)

“Smith induced random mutations in the genome of S. gallinarum SG9 strain, resulting in the development of the commercial strain SG9R. The efficacy of SG9R was demonstrated in preventing fowl typhoid during challenge experiments. However, the SG9R strain still exhibited residual pathogenicity and caused fowl typhoid symptoms in vaccinated birds. Additionally, the presence of mutation sites in the aceE and rafJ genes within the SG9R strain's genome raises potential concerns regarding reversion. Additionally, the close relationship between the SG9R strain and S. gallinarum isolates from a fowl typhoid outbreak in Belgium has been elucidated through whole-genome sequencing, highlighting significant safety considerations associated with the SG9R strain. Consequently, there is an increased demand for S. gallinarum vaccine candidates that exhibit enhanced safety profiles suitable for clinical application.” (Lines 81-92)

Comments 6: Introduction section. Information on the current status of commercial used vaccine or control strategies against such infections should be reported.

Response 6: Thank you. We consider this review comment to be similar to Comment No. 5, so we have responded uniformly under Comment No. 5, which includes information on commercial vaccines.

Comments 7: Materials and Methods section. Information on method of blood collection and serum preparation should be added.

Response 7: Agree. We have added the information on method of blood collection and serum preparation in the revised manuscript. “Blood samples were collected from a fixed number of 10 chickens at 7, 14, 21, and 28 days after the first immunization via the wing vein. Serum samples were obtained by centrifuging blood samples at 4000 rpm for 5 minutes as test samples.” (Lines 224-226)

Comments 8: Materials and Methods section. Lines 128, 174, 175 have words with different font size.

Response 8: Thank you for pointing these out. We have modified their font sizes. (Line 145 and 191)

Comments 9: Materials and Methods section. Regarding vaccination regime, how many trials was conducted and if only one trial, the authors should defend this point to guarantee the reproducibility.

Response 9: Thank you for pointing this out. The reproducibility of this study can be guaranteed. As a matter of fact, we conducted three comprehensive evaluations of the vaccine candidates over a span of two years to assess their protective efficacy.

Comments 10: Discussion section. The section of discussion is very poor and need addition of more information on the importance of developing vaccines against such infections, mechanism of induced immunoprotection, superiority of SG102 as a vaccine candidate, utility of use against both APEC O78 and O161 serogroups as well as Salmonella gallinarum infections, prospects to improve the survival of chicken in the future studies, efficacy against in case of different type of chickens…

Response 10: Agree. We have added the relevant contents in the revised manuscript. “The significant threat posed by APEC to global poultry production results in substantial economic losses, highlighting the imperative of developing vaccines that can provide robust protection against a broad spectrum of APEC strains.” (Lines 520-522)

“Less-virulent Salmonella has been shown to effectively induce robust humoral, mucosal, and cell-mediated immune responses as a live vector. The target antigen can be stably expressed by inserting it into an expression plasmid that carries the complementary nutritional deficiency gene of the bacterial vector based on the chromosome-plasmid balanced lethal system.” (Lines 523-527)

“A comprehensive understanding of the mechanism underlying induced immune protection is pivotal for assessing the efficacy of vaccine candidates. Type I fimbria is biosynthesized by APEC strains and is a critical virulence determinant involved in adherence to eukaryotic cells. FimH, the adhesin portion of type I fimbriae, mediates the adherence to host cells through specific combinations with mannose receptors (MR) and the adherence can be inhibited by the D-mannose. In addition to playing an essential role in host adherence, FimH could activate innate immune responses in vitro and in vivo. Furthermore, FimH-mediated mucosal adjuvant effects on the induction of specific immune responses were determined via intranasal administration.” (Lines 540-548)

“In this study, by expressing and displaying APEC type I fimbriae on the surface of the vector strain, the in vivo persistence time of the SG102 strain in the liver, spleen, and cecum from SG102-immunized chickens was at least 2 weeks, which is longer than that of the SG101 strain. Bacterial loads of the SG102 strain in livers, spleens and ceca were significantly higher than those of the SG101 strain (Figure 4C and D). The prolonged persistence of SG102 is a distinctive feature that enhances its colonization levels and adherence capabilities, thereby contributing to heightened protective effects.”(Lines 554-560)

“Further studies are necessary to evaluate the protective effects of the vaccine candidate against other dominant APEC serogroups such as O1 and O2 in flocks of varying breeds. In addition, the protective efficacy of SG102 against APEC O78 and O161 serogroup infections should be further enhanced in subsequent studies to optimize its clinical applicability.” (Lines 597-601)

4. Response to Comments on the Quality of English Language.

Point 1: Minor editing of English language required.

Response 1: Thank you for pointing this out. We worked hard to do the revision for the manuscript.

Reviewer 2 Report

Comments and Suggestions for Authors

The comments were implemented into the text. 

Author Response

Response to Reviewer 2 Comments

  1. Summary

Thank you for dedicating your valuable time to reviewing this manuscript. Please find the detailed responses below and the corresponding revisions/corrections highlighted/in track changes in the re-submitted files.

2. Questions for General Evaluation

Reviewer’s Evaluation

Does the introduction provide sufficient background

and include all relevant references?

Can be improved

Are all the cited references relevant to the research?

Yes

Is the research design appropriate?

Can be improved

Are the methods adequately described?

Yes

Are the results clearly presented?

Yes

Are the conclusions supported by the results?

Yes

  1. Point-by-point response to Comments and Suggestions

The comments were implemented into the text (original manuscript). We copied the whole comments from the text and responded accordingly.

Comments 1: ‘Cloning of APEC Type I Fimbriae Operon and Construction of Chromosome-Plasmid Balance Lethal System of SG01 asd Deletion Mutant’ (line 116-126). This section must explain more in details. Please explain the process of stable gene integration in chromosome using Red-recombination method.

Response 1: Agree. We have added the process of stable gene integration in chromosome using Red-recombination method in the revised manuscript.

“The primers P3 and P4 are used to identify the asd gene. Homologous recombinant primers (P1-C1 and P2-C2) are designed on the medial side of P3 and P4, in which the sequences displayed in lowercase letters are homologous to the asd gene of the SG01 strain, and the sequences displayed in uppercase letters are homologous to the cat gene of the pKD3 plasmid. The homologous recombination fragment containing the cat gene was amplified using the pKD3 plasmid as a template. Initially, the pKD46 plasmid was transformed into SG01 to construct the recombinant strain named SG01 (pKD46), which contained the pKD46 plasmid. The expression of the recombinase is induced by the addition of L-arabinose during the preparation of competent cells for the SG01 (pKD46) strain. Then, the homologous recombination fragment containing the cat gene was transformed into the SG01 (pKD46) strain to replace the asd gene segment with the cat gene segment, and the recombinant strain SG01Δasd::cat was constructed. The pCP20 plasmid, which expresses FLP flipase to bind to the FRT sites at both ends of the cat gene segment, is transformed into the SG01Δasd::cat strain. Ultimately, the SG01 strain lacking the asd gene segment, named SG100, was constructed.” (Lines 119-134)

Comments 2: Please indicate clearly the differences between these recombinant strains (SG01, SG101, SG102 and SG103) and also about E. coli strains.

Response 2: Thank you for pointing this out. We have indicated clearly the differences between these recombinant strains (SG01, SG101, SG102 and SG103) and also about E. coli strains in ‘Table 1 (Strains and plasmids used in this study)’ in the manuscript.

Comments 3: Line 174-175. Please correct the font.

Response 3: Agree. We have modified their font sizes in the revised manuscript (Line 191).

Comments 4: Figure 1 (line 278). I recommend to add a figure containing the schematic representation for deletion process.

Response 4: Agree. We have added the schematic representation for deletion process in Figure 1 (Figure 1D) in the revised manuscript.

Comments 5: Figure S2 (line 287). This figure explains important results and it should be added into the main text. The figure legend needs more exact explanation and addressing.

Response 5: Thank you for pointing this out. We have added the figure S2 with the exact figure legend into the main text in the revised manuscript (Figure 2; Line 354-391).

  1. Response to Comments on the Quality of English Language.

Point 1: I am not qualified to assess the quality of English in this paper.

Response 1: Thank you for pointing this out. We worked hard to do the revision for the manuscript.

Reviewer 3 Report

Comments and Suggestions for Authors

In their article” Recombinant S. Gallinarum vaccine candidate expressing avian pathogenic Escherichia coli type I fimbriae provides protections against APEC O78 and O161 serogroups and S. Gallinarum infection “ the authors present a study, whose aim is to present a vaccine candidate against Avian pathogenic Escherichia coli (APEC) O78 and O161 serogroups, and the vector S. Gallinarum. The recombinant strain SG102 was developed by expressing the APEC type I fimbriae gene cluster (fim) on the cell surface of avirulent Salmonella Gallinarum (S. Gallinarum) vector strain using a chromosome-plasmid balanced lethal system.

The information in the manuscript is relevant and interesting. Escherichia coli infections are widely distributed among poultry of all ages and categories and cause devastating economic losses to the industry.  

I have some minor points for the author to consider:

1.                  Materials and Methods. Page 3 “Bacterial Strains, Plasmids, and Growth Conditions” There is no information about growth conditions for some of the strains and plasmids included in Table 1.

2.                  Line 112 “purchased” is better to be replaced by “supplied”

3.                  Page 4. Line 157. Page 5. Line 171. You need to cite at the end of the paragraphs.

4.                  Results. Figures 3, 5, S1 and S3. It would help if you revised the text. It is too long and more proper for the Materials and Methods, or for the Results. You need brief, clear information about exactly what is seen in the figure, not what you did.

5.                  Avoid the word “briefly”, it is clear that the method is described briefly.

Author Response

Response to Reviewer 3 Comments

  1. Summary

Thank you for dedicating your valuable time to reviewing this manuscript. Please find the detailed responses below and the corresponding revisions/corrections highlighted/in track changes in the re-submitted files.

2. Questions for General Evaluation

Reviewer’s Evaluation

Does the introduction provide sufficient background

and include all relevant references?

Yes

Are all the cited references relevant to the research?

Can be improved

Is the research design appropriate?

Yes

Are the methods adequately described?

Yes

Are the results clearly presented?

Yes

Are the conclusions supported by the results?

Yes

  1. Point-by-point response to Comments and Suggestions

The comments were implemented into the text (original manuscript). We copied the whole comments from the text and responded accordingly.

Comments 1: Materials and Methods. Page 3 “Bacterial Strains, Plasmids, and Growth Conditions” There is no information about growth conditions for some of the strains and plasmids included in Table 1.

Response 1: Agree. We have added the information about growth conditions for strains and plasmids in Table 1 in the revised manuscript.

Comments 2: Line 112 “purchased” is better to be replaced by “supplied”.

Response 2: Thank you for pointing this out. We have replaced ‘purchased’ by ‘supplied’ in the revised manuscript (line 113 and 114).

Comments 3: Page 4. Line 157. Page 5. Line 171. You need to cite at the end of the paragraphs.

Response 3: Agree. We have added the relevant citations in the revised manuscript (line 174 and 189).

Comments 4: Results. Figures 3, 5, S1 and S3. It would help if you revised the text. It is too long and more proper for the Materials and Methods, or for the Results. You need brief, clear information about exactly what is seen in the figure, not what you did.  

Response 4: Thank you for pointing this out. The figure legends are relatively long because the first image in each of them represents a flow chart, which includes the vaccination of the vaccine strain, challenge, and the collection times for the samples. Considering that this may help readers quickly understand relevant information when viewing the results in the figures, we have retained these details.

Comments 5: Avoid the word “briefly”, it is clear that the method is described briefly.

Response 5: Agree. We have deleted the word ‘briefly’ in the revised manuscript.

  1. Response to Comments on the Quality of English Language.

Point 1: I am not qualified to assess the quality of English in this paper.

Response 1: Thank you for pointing this out. We worked hard to do the revision for the manuscript.

Reviewer 4 Report

Comments and Suggestions for Authors

This is an interesting manuscript since APEC is causing death and substantial economic loss in the poultry industry. In this study, the recombinant strain SG102 was developed by expressing the APEC type I fimbriae gene cluster (fim) on the cell surface of avirulent Salmonella Gallinarum (S. Gallinarum) vector strain using a chromosome-plasmid balanced lethal system. The expression of APEC type I fimbriae was verified.  At two weeks after oral immunization, the SG102 strain remained detectable in the livers, spleens, and ceca of SG102-immunized chickens. At 14 days after the secondary immunization with 5×109 CFU of the SG102 strain orally, highly antigen-specific humoral and mucosal immune responses against APEC type I fimbriae protein were detected in SG102-immunized chickens, with IgG and secretory IgA (sIgA) concentrations of 221.50 μg/mL and 1.68 μg/mL, respectively. The survival rates of SG102-immunized chickens were 65% (13/20) and 60% (12/20) after challenge with 50 LD50 doses of APEC virulent strains O78 and O161 serogroups. The protection rate is of importance but of a low rate, therefor this SG102 vaccine has of medium chance to be a good vaccine used widely.

Author Response

Response to Reviewer 4 Comments

  1. Summary

Thank you for dedicating your valuable time to reviewing this manuscript.

2. Questions for General Evaluation

Reviewer’s Evaluation

Does the introduction provide sufficient background

and include all relevant references?

Yes

Are all the cited references relevant to the research?

Yes

Is the research design appropriate?

Yes

Are the methods adequately described?

Yes

Are the results clearly presented?

Yes

Are the conclusions supported by the results?

Can be improved

  1. Point-by-point response to Comments and Suggestions

Comments 1:

This manuscript is dealing with production of a promising vaccine candidate against APEC O78 and O161 serogroups as well as S. Gallinarum infections. The vaccine against S. Gallinarim was found to be an effective vaccine, by adding the APEC O78 and O161 antigens it should provide a protection against these two APEC which are causing a substantial loss to the poultry industry.

The introduction is covering the known information, the methods are proper and providing wide and comprehensive results. The references are appropriate.

A recombinant vaccine candidate, derived from S. Gallinarum O1, expressing APEC O78 and O161 type I fimbriae for adhering abilities in vitro and in vivo. The immune responses and protective effects of the vaccine candidate against APEC O78 and O161 serogroups, as well as S. Gallinarum infection, were evaluated in several aspects including challenge studies.

In the APEC challenge assay, 65% of the present produced vaccine SG102-immunized chickens (13/20) survived during the observation period, which was significantly high than the control group. Sixty five percent protection in experimental condition is considered acceptable but it is a low protection percentage in field condition.

APEC O78 and O161 are only two of several APEC strains pathogenic and disease causing in the poultry industry and therefor the authors summing up this manuscript by stating “Further studies are necessary to evaluate the protective effects of the vaccine candidate against other dominant APEC serogroups such as O1 and O2”.

Since this is a preliminary study with moderate protective-vaccine results, it has some interest from the scientific and experimental side but cooperatively low interest from the practical aspect.

Response 1: Thank you for agreeing with the research methodology, results, and references. For the point of view “Sixty-five percent protection in experimental condition is considered acceptable but it is a low protection percentage in field condition”, we agree with it to some extent. In this study, we inoculated APEC pathogenic strain O78 with a dose of 50 LD50 by the posterior chest air sac route, which is a highly pathogenic route or dosage. Under this dosage, 65% of SG102-immunized chickens survived at this dose. It is worth noting that non-lethal APEC infections also occur in clinical fields. In such cases, immunization with the SG102 strain in chickens may exhibit higher protective effects. Our lab will further optimize SG102 strain by displaying other APEC protective antigens on the surface of Salmonella vector through fusion expression with APEC type I fimbriae. We hope to increase the protective efficacy of the recombinant strain against APEC infection in further studies.

  1. Response to Comments on the Quality of English Language.

Point 1: I am not qualified to assess the quality of English in this paper.

Response 1: Thank you for pointing this out. We worked hard to do the revision for the manuscript.

Round 2

Reviewer 1 Report

Comments and Suggestions for Authors

The authors have been responded to the comments and the quality of the manuscript has been improved significantly.

I just have one comment, in title, it is better to use Salmonella gallinarum in full name in first appearance then use S. gallinarum in the second appearance